# Identification of Nordic Berries with Beneficial Effects on Cognitive Outcomes and Gut Microbiota in High-Fat-Fed Middle-Aged C57BL/6J Mice

**DOI:** 10.3390/nu14132734

**Published:** 2022-06-30

**Authors:** Fang Huang, Nittaya Marungruang, Olha Kostiuchenko, Nadiia Kravchenko, Stephen Burleigh, Olena Prykhodko, Frida Fåk Hållenius, Lovisa Heyman-Lindén

**Affiliations:** 1Division of Biotechnology, Department of Chemistry, Lund University, 221 00 Lund, Sweden; 2Aventure AB, Scheelevägen 22, 223 63 Lund, Sweden; 3Berry Lab AB, Scheelevägen 22, 223 63 Lund, Sweden; nittaya.marungruang@aventureab.com (N.M.); lovisa.heyman.linden@aventureab.com (L.H.-L.); 4Department of Food Technology, Engineering and Nutrition, Lund University, 221 00 Lund, Sweden; kostiuchenko.olha@biph.kiev.ua (O.K.); kravchenko.nadiia@biph.kiev.ua (N.K.); stephen.burleigh@food.lth.se (S.B.); olena.prykhodko@food.lth.se (O.P.); frida.hallenius@food.lth.se (F.F.H.); 5Department of Cytology, Bogomoletz Institute of Physiology, 010 24 Kyiv, Ukraine; 6Department of Experimental Medical Science, Lund University, 221 84 Lund, Sweden

**Keywords:** berries, high-fat diet, cognitive function, gut microbiota, hippocampal neurogenesis

## Abstract

High-fat diets are associated with neuronal and memory dysfunction. Berries may be useful in improving age-related memory deficits in humans, as well as in mice receiving high-fat diets. Emerging research has also demonstrated that brain health and cognitive function may be related to the dynamic changes in the gut microbiota. In this study, the impact of Nordic berries on the brain and the gut microbiota was investigated in middle-aged C57BL/6J mice. The mice were fed high-fat diets (60%E fat) supplemented with freeze-dried powder (6% dwb) of bilberry, lingonberry, cloudberry, blueberry, blackcurrant, and sea buckthorn for 4 months. The results suggest that supplementation with bilberry, blackcurrant, blueberry, lingonberry, and (to some extent) cloudberry has beneficial effects on spatial cognition, as seen by the enhanced performance following the T-maze alternation test, as well as a greater proportion of DCX-expressing cells with prolongation in hippocampus. Furthermore, the proportion of the mucosa-associated symbiotic bacteria *Akkermansia muciniphila* increased by 4–14 times in the cecal microbiota of mice fed diets supplemented with lingonberry, bilberry, sea buckthorn, and blueberry. These findings demonstrate the potential of Nordic berries to preserve memory and cognitive function, and to induce alterations of the gut microbiota composition.

## 1. Introduction

In the coming decades, it is projected that one in six people will be over age 65 (16%), and that the number of people 80 years or over is expected to triple, from 143 million in 2019 to 426 million by 2050 [1].

An aging world population will also lead to increased prevalence of cognition-related diseases. To date, there are no effective medical cures to prevent the decline in cognitive functions of the aging brain. Processes that lead to losses in cognitive function often occur gradually over the course of decades and may ultimately lead to the development of mild cognitive impairment or the onset of neurodegenerative diseases such as dementia [2]. It is of great interest to investigate preventive strategies to limit cognitive decline; hence, the identification of novel foods with cognitive benefits is an important approach to promote healthy aging in the population.

Benefits from eating certain diets and foods have gained interest in the field of brain research [3,4]. Studies in both animal models and humans have shown that certain berries can improve performance in cognitive tasks and aid in improving age-related decline in memory and learning [5]. The underlying mechanisms of such improvement following the consumption of berries remain to be elucidated. However, recent evidence indicates that the consumption of foods rich in dietary fiber and polyphenols, such as berries, may exert actions partly via the gut–brain axis [6].

Several studies have used high-fat (HF) feeding to induce cognitive deficits in rodents to model human cognitive impairment [7,8,9]. American blueberries have been studied in this setting and are shown to alleviate HF-induced cognitive impairment in mice [7,8]. However, there are many types of edible berries which remain largely unexplored in terms of effects on the brain and memory. Nordic berries, rich in polyphenolic and other potentially bioactive compounds, have also been shown to confer health benefits, including improvements in cardiometabolic parameters as well as anti-inflammatory effects [10].

The aim of this study was to explore the effects of consuming different Nordic berry varieties, which have thus far not been investigated in vivo, on cognitive function, related biomarkers, and associated changes in gut microbiota composition. The study was conducted using high-fat-fed middle-aged C57BL/6J mice (24 weeks old) as a model for age- and lifestyle-induced cognitive impairment. 

## 2. Materials and Methods

### 2.1. Experimental Diets

Five different Nordic berries, including lingonberry *Vaccinium vitis-idaea*, bilberry *Vaccinium myrtillus*, cloudberry *Rubus chamaemorus* (Olle Svenssons Partiaffär AB, Olofström, Sweden), blackcurrant *Ribes nigrum* (SLU Balsgård Kristianstad, Sweden), and sea buckthorn *Hippophae rhamnoides* Sol cultivar (SLU Balsgård Kristianstad/Wrams skafferi Tollarp, Sweden), as well as “American” blueberry *Vaccinium ashei* Tifblue cultivar (Folsom, CA, USA), were used in this study. The Nordic berries (all except blueberry) were freeze-dried at SLU Balsgård, Kristianstad, Sweden. The freeze-dried Nordic berries were then ground into fine powder using a kitchen blender. Blueberry was provided as freeze-dried powder in de-oxygenated cans. All the berry powders were flushed with nitrogen gas, vacuum-packed in bags with oxygen absorbers, and stored at room temperature until being incorporated into rodent diet pellets.

All diets were formulated by Research Diets, Inc. (New Brunswick, NJ, USA) with the inclusion of essential macro- and micronutrients for rodents. A total of eight experimental diets were formulated, including a low-fat (LF) control diet (10% calories from fat, modified from D12450J); a high-fat control diet (60% calories from fat, modified from D12492); and HF diets supplemented with each of the berry powders at a dose of 6% (*w*/*w*) on a dry weight basis. All HF diets were matched on macro- and micronutrients, as well as sucrose, fructose, and glucose, i.e., adjusted for the amounts coming from 6% of the freeze-dried berry powders added (Appendix A). Diets were stored in bags flushed with nitrogen gas and stored at −20 °C until use. 

### 2.2. Animal Experiment

One hundred and thirty-five male C57Bl/6J mice (Janvier-Labs, Le Genest-Saint-Isle, France) arrived at the animal facility at the age of 22 weeks and were given standard chow (RM1, SDS) ad libitum for 2 weeks for acclimatization (22 °C, 12 h light–dark cycle). Subsequently, the mice were allocated into weight-matched groups (initial mean mouse weight was 29 g) of 15 animals (5 animals/cage) and randomly assigned to receiving one of the eight diets for 4 months. 

Animals and food were weighed, and food was replaced every week. The food intake per cage was measured weekly, with 3 cages (*n* = 5/cage) per diet group. However, due to the deaths of unknown causes of some individuals (*n* = 1 from LF, *n* = 1 from HF, *n* = 1 from the blackcurrant group) and a situation where the cages were split into sub-cages to avoid fighting, weekly food intake was calculated and presented as the average consumption per mouse instead of per cage. The study was approved by the Local Animal Experiment Ethical Review Committee in Lund, Sweden (approval number 5.8.18—13,983/2018). 

### 2.3. Cognitive Behavior Evaluation

T-maze spontaneous alternation and novel object recognition (NOR) tests were conducted to assess the effects of the diets on spatial and recognition memory of the mice after a 15— and 16—week feeding period, respectively. Monitoring was performed with a camera system using the Ethovision XT 14.0 software (Noldus Information Technology b.v., Wageningen, The Netherlands) for both video tracking and video analysis. The behavior tests were performed during the light phase.

#### 2.3.1. T-Maze Spontaneous Alternation 

The protocols of the test were published with various modifications. In this study, a discrete-trail-without-reward protocol was adapted from two published procedures with minor modifications [11,12]. The T-maze was made of transparent acrylic and consisted of three arms (two goal arms: 30 cm × 10 cm; one starting arm: 30 cm × 10 cm). A complete trial consisted of two phases; in the first phase, the mouse was put at the starting arm for 10 s before the guillotine door opened, allowing it to choose between one of the two goal arms. Once the mouse had chosen an arm, the guillotine door was shut to keep the mouse in the chosen arm for 30 s. For the second phase, the mouse was returned to the starting position and allowed to spontaneously choose one of the goal arms again. A total of two trials were carried out per mouse per day—one in the morning and the other in the afternoon. Fresh woodchip bedding was put in the maze and changed between each cage. The number of explorations in each arm was measured. The alternation rate (%) was evaluated as the ratio of correct arm choices (unvisited goal arm in the second phase) to the total number of trials, multiplied by 100. The protocol was based on the tendency of animals to explore the novel arm rather than the one previously explored without rewarding stimuli, called ‘spontaneous alternation’. A rate higher than 50% may reflect a preference for choosing a novel arm, meaning that the animal would have a memory of the location of the arm which was already visited [13].

#### 2.3.2. Novel Object Recognition (NOR) Test 

A novel object recognition (NOR) protocol, previously described [14], was followed with minor modifications. The apparatus consisted of a white rectangular open field arena (38 cm × 50 cm × 30 cm). The experiment consisted of three sessions: habituation, familiarization, and the test session. During the habituation session, mice were transferred from the cage to the arenas in the absence of objects and allowed to freely explore the empty arena for 10 min. During the familiarization session, two identical objects (two 1.5 mL Eppendorf tubes or two binder clips, approximately 1.5 cm in diameter) were placed in the arena, and the mice were given 10 min to freely explore the arena with the two identical objects. The positions of the two objects (position A and B) were symmetric and fixed between mice to avoid innate preference for a location. The test session was performed after a 3–4 h retention time, where one of the two objects (object in position B) was replaced by a novel item, e.g., one tube was replaced with a clip if two tubes were presented in the familiarization session. The mice were then allowed to explore the new configuration for 10 min. In between sessions, the mice were placed back to their home cages, and the arenas and objects were wiped off with mild soap and 10% ethanol to remove any odor. The mice were considered to have explored the objects when they were facing or sniffing with their nose directly on the objects or within the area of 1.5 cm around the objects. Approaching the objects with other parts of the body or sitting on them was not considered as exploring. The percentage of time spent exploring the object in position B relative to time spent around both objects from both familiarization and test sessions was calculated to reveal preferences for the novel object. Position B was a term employed in video analysis, which refers to the position where the novel object was placed, considering that the novel object position was randomly chosen between one of the two positions to avoid any orientation preference of the animals. Preference for the novel object was expressed as the time spent exploring the object in position B in the test session in comparison to the time spent exploring the object in the same position in the familiarization session (per cent time spent on object in position B). The discrimination index (DI) was computed as the time exploring the new object divided by the total time exploring both objects in the test session.

### 2.4. Sample Collection 

At the end of the study, 4 h fasted animals were anesthetized using isoflurane (Abbott, Chicago, IL, USA). Blood samples were collected from the heart before the mice were terminated and the brains were quickly dissected. Organ weights were recorded, and blood plasma samples were separated via centrifugation at 2000× *g* for 10 min. All tissues were snap-frozen and subsequently stored at −80 °C until further analyses.

### 2.5. Brain Dissection

Hippocampi from both left and right hemispheres were dissected and stored at −80 °C for biochemistry analyses. Three brains per group were randomly selected for histology analyses. For these brains, the right hemispheres (*n* = 3 per group) were carefully dissected and fixed in 4% paraformaldehyde in phosphate buffered saline (PBS) for 72 h at 4 °C before being rinsed and incubated in PBS for 24 h at 4 °C. The hemispheres were then stored in PBS with 0.02% NaN_3_ at 4 °C until further cryoprotection steps for histology analyses.

### 2.6. Hippocampus Protein Analyses 

Hippocampus homogenates were prepared from both the left and right hemisphere (*n* = 10−12/group), or only the left hemisphere from those mice randomly assigned for histology analyses (*n* = 3/group). The tissues were homogenized on ice in Tris-buffered saline with 1% Triton X100 and 1% protease inhibitor cocktail using an electric homogenizer. The homogenates were maintained on constant agitation for 2 h at 4 °C and centrifuged for 20 min at 13,000 rpm. Supernatants were collected and the total protein concentrations in the homogenates were evaluated using the Pierce™ BCA Protein Assay Kit (Pierce Biotechnology, Illinois, IL, USA) according to the manufacturer’s protocol.

Levels of brain-derived neurotrophic factor (BDNF) (Human BDNF SimpleStep ELISA^®^ kit, Abcam PLC, Cambridge, UK) and interleukin (IL)–1β (Mouse IL–1 beta Uncoated ELISA kit, Thermo Fisher Scientific Inc., Waltham, MA, USA) in hippocampus homogenates were determined by enzyme-linked immunosorbent assays according to the manufacturer’s protocol. BDNF results were reported per mg protein in the brain and the brain homogenates were made on different weights of hippocampus.

### 2.7. Brain Immunohistochemistry

The hippocampal area was dissected from one hemisphere, placed in the acrylic coronal brain matrix (*n* = 3 brains per group), cryoprotected by immersion into 30% sucrose in 1× PBS overnight, and then put into 50% sucrose solution in 1× PBS overnight. The samples were embedded into cryomolds (Sakura TissueTek^®^ cryomold, 25 × 20 mm) using OCT cryomount (Histolab OCT) and then frozen on a metal cylinder in liquid nitrogen. Then, 10 µm sections between coordinates Bregma −1.34 mm and −2.18 mm [15] were cut serially using a Leica CM1860 Cryostat (Leica Microsystems AB, Wetzlar, Germany) and then were mounted on a polylysine-coated adhesion slide (ThermoFisher Scientific), dried for 1 h at room temperature, and stored at −20 °C. Morphological features of the hippocampus were assessed using hematoxylin and eosin staining. Then, 2–10 slides from each group were used for immunohistochemistry. Sections were subjected to antigen retrieval (50 mM Tris–HCl buffer with 0.05% Tween 20 (pH 9.0) and heating for 20 min at 85–95 °C), before then being incubated in the anti-Doublecortin (anti-DCX) (E–6; sc–271390; Santa Cruz Biotechnology, Dallas, TX, USA) mouse primary antibody conjugated with Alexa Fluor 546 at 1:100 dilution in 1% BSA blocking buffer for 2 h at room temperature and subsequently mounted with Fluoromount G™ [5]. Images were taken using an Olympus microscope coupled to a digital camera and illuminated with a fluorescent light source. Quantitative analysis was performed within the Dentate Gyrus (DG) region using ImageJ software (ImageJ, U. S. National Institutes of Health, Bethesda, MD, USA). The regions of interest were the granule cell layer (GCL) and the sub-granular cell layer (SGL). On each section, Doublecortin (DCX)-expressing cells in the granule and subgranular layers of the DG were counted by an investigator blinded to the group assignments at 400× magnification. Additionally, DCX-expressing cells with prolongations were quantified. Parallel to DCX-expressing cells counting, the %DCX+ area in DG was assessed at 100× magnification. To exclude autofluorescence negative control, images were assessed, and its values were subtracted from DCX images values. For quantitative evaluation of cells, 6–9 images from each section were used, whereas 3 images were used for %DCX+ area assessment. Results are expressed as the total number of DCX-expressing cells in the whole DG, the percentage of DCX-expressing cells with dendrites, and the percentage of the DCX+ area in DG.

### 2.8. Cecal Microbiota 

DNA from the cecal tissue and content was extracted using the QIAamp DNA Stool Mini Kit (Qiagen, Hilden, Germany), with the addition of a bead beating step. The V4 region of 16S rRNA genes was amplified using forward and reverse primers containing Illumina overhang adaptors and unique dual indexes. The sequence of the 16S amplicon primers [16] was (forward) –5′ TTGCCAGCMGCCGCGGTAA and (reverse) –5′ GGACTACHVGGGTWTCTAAT. Paired-end sequencing with a read length of 2 × 250 bp using the Miseq V2 Reagent Kit was carried out on a Miseq Instrument (Illumina Inc., San Diego, CA, USA). Sequencing data were analyzed using the open-source bioinformatics pipeline Quantitative Insights into Microbial Ecology (QIIME) [17]. The sequences were grouped into operational taxonomic units (OTUs) by UCLUST at a minimum of 97% sequence similarity. Representative sequences (most abundant) from each OTU were aligned using python nearest alignment space termination (PyNAST) [18]. Taxonomy was assigned using the Greengenes database (v.13.8) [19].

### 2.9. Statistical Analyses 

Unless otherwise stated, data are displayed as mean ± SD and analyzed by one-way ANOVA, followed by Dunnett’s test for multiple comparisons versus the HF control group. In cases where normality of the data could not be assumed, groups were compared with the non-parametric Kruskal–Wallis test followed by Dunn’s post-hoc test. In T-maze data, the non-parametric Wilcoxon test was applied to compare the alternation rate of each group against a 50% chance. In the NOR data, the non-parametric Wilcoxon signed-rank test was used to compare the percentage time spent exploring object on Position B in the familiarization sessions to the test sessions in order to identify the preference for the novel object in different diets groups. A one-way ANOVA was used to compare DI in the test session between HF control with the berry groups. The histological part was considered exploratory due to the low number of animal brains that could be collected; therefore, the quantitative data on DCX analyses were reported without statistical implication.

For the 16S data, the diversities analyzed using the Shannon index from each group were compared to the HF control group using the Kruskal–Wallis rank-sum test, followed by pairwise comparisons using the Wilcoxon rank-sum test. The unique observed species and the total OTU richness were compared using the ANOVA test, followed by the least significant difference (LSD) post-hoc test when ANOVA indicated significance. The OTU data from each group were compared to the HF control group using two-way ANOVA and the *p*-values were corrected for multiple comparisons by controlling the false discovery rate (FDR) using the original FDR method of Benjamini and Hochberg. If the data were normally distributed, data were presented as means in bar charts; otherwise, the data were presented as median values in box plots.

GraphPad Prism version 8.01 (GraphPad Software, Inc., San Diego, CA, USA) was used for statistical analyses. 

## 3. Results

### 3.1. Food Intake and Body Weight

No significant differences in the food intake were observed among groups receiving HF diets with or without berries, whereas the LF group consumed significant higher amount of food in weeks 5, 6, and 9 (all *p* < 0.05) compared to the HF control group (Figure 1a). 

In addition, no significant differences were seen in the energy intake (kcal) among the groups receiving HF with or without berries, whereas the LF group had slightly (*p* < 0.05) lower energy intake compared to HF diets in weeks 1, 5, 7, 8, 9, and 12 (data not shown).

Compared to the HF control group, the body weight gain (*p* < 0.001) and the epididymal fat pad weight (*p* < 0.001) were significantly lower in mice receiving the LF diet (Figure 1b,c), after 4 months on diets. 

There were no significant differences in the body weight gain and fat pad weight of the mice fed with a HF diet supplemented with different berries when compared to the HF control group (Figure 1b,c), yet a non-significant trend for smaller epididymal fat pads in the HF group supplemented with bilberries was observed (*p* = 0.076). 

### 3.2. Effects on Spatial Memory Performance in T-Maze Test 

The T-maze test revealed berry-specific differences in the ability of mice to perform in a spatial memory task. The results show that the observed alternation rate between maze arms in the HF control group was not significantly different (*p* = 0.125) from a random chance 50% alternation rate (Figure 2). Mice fed with the LF diet showed an average alternation rate of 71%, which tends to be higher (*p* = 0.07) compared to the 50% random chance rate. The groups fed with the HF diet supplemented with bilberries (alternation rate: 83%, *p* < 0.01), blackcurrant (alternation rate: 79%, *p* < 0.01), blueberry (alternation rate: 79%, *p* < 0.05), as well as lingonberry and cloudberry (both alternation rate 73%, *p* < 0.05) showed average alternation rates that are all significantly higher than a 50% random chance (Figure 2). There was no statistically significant effect reported, but a trend was observed in the sea buckthorn group (*p* = 0.727, 57% alternation rate). 

### 3.3. Effects on Recognition Memory in the NOR Test

The LF group, as well as mice receiving the HF diet supplemented with cloudberries, spent significantly more time on the Position B (novel object) in the test session compared to the time spent on the object located in the same position in the familiarization session (Figure 3).

In the remaining HF groups, there was a tendency of higher novel object exploration amongst mice receiving HF alone (63%, *p* < 0.1), as well as HF supplemented with blackcurrant (59%, *p* < 0.1) or sea buckthorn (58%, *p* < 0.1). No statistically significant differences in object exploration were observed in the lingonberry, bilberry, or blueberry groups. DI in the test session showed no significant difference among the diets (Figure 3b).

### 3.4. Effects on Hippocampal Neurogenesis and Expression of Brain-Derived Neurotrophic Factor (BDNF) 

The expression of doublecortin (DCX) protein in the dentate gyrus (DG) was detected in (a) the soma of cells located in the sub-granular layer (SGL); (b) its prolongations in the granule cell layer (GCL) and molecular layer; and (c) some hilar interneurons (Figure 4a).

DCX-expressing cells were classified into two categories according to the presence and the shape of dendrites (Figure 4b–i). Category 1 were cells with no or short processes (indicated by purple arrowheads), whereas Category 2 were cells with long processes and a more mature appearance (indicated by white arrowheads). 

DCX-expressing Category 1 cells were more common in the brain of mice receiving LF, HF, HF + cloudberry, and HF + sea buckthorn diets, whereas Category 2 cells were widespread in the DG of mice fed with the diet supplemented with lingonberry, bilberry, blackcurrant, and blueberry. Mice fed with the LF and HF diet without berries displayed relatively faint DCX expression in the dendritic tree, branching into the molecular layer, as opposed to the mice fed with the HF diet supplemented with lingonberry, bilberry, or blueberry (Figure 4b–i).

Quantification of the images indicated that higher levels of DCX-expressing cells (Figure 4j) and a higher %DCX area (Figure 4k) were present in groups supplemented with lingonberry, bilberry, or blackcurrant. 

There were no differences in hippocampal protein concentration of the brain-derived neurotrophic factor (BDNF) in mice receiving different diets (Appendix A). Similarly, the levels of cytokine IL-1β in hippocampus remained unchanged between groups (data not shown). In addition, amyloid-beta staining was performed but showed no deposits in the hippocampus of animals in any diet group in the current study (data not shown).

### 3.5. Cecal Microbiota

The analysis of cecal bacterial 16S rRNA genes revealed that the Shannon diversity indices, a metric that weights the numbers of species by their relative evenness data, were significantly higher in groups supplemented with lingonberry (*p* < 0.001), blackcurrant (*p* < 0.001), cloudberry (*p* < 0.01), sea buckthorn (*p* < 0.001), and blueberry (*p* < 0.01) as compared to the HF control group (Figure 5a). The total OTU richness, i.e., the total number of OTUs or species recorded, showed that the HF group had lower richness as compared to other groups (*p* < 0.001), while the unique observed species were mostly observed in the LF group (*p* < 0.05) (Figure 5b,c).

Four-month feeding with the experimental diets induced drastic changes in the cecal microbiota profiles. At the phylum level, the cecal microbiota in mice fed with the HF control diet were dominated by Firmicutes (58%) and Proteobacteria (23%), Bacteroidetes (19%), and Verrucomicrobia (1%) (Figure 6a). As compared to the HF group, Firmicutes (58% in the HF group) decreased to 46% (*p* < 0.0001) in the group supplemented with bilberry, and increased in groups supplemented with blackcurrant, cloudberry, sea buckthorn, and blueberry to 66% (*p* < 0.001), 66% (*p* < 0.001), 62% (*p* < 0.01), and 65% (*p* < 0.001), respectively. Proteobacteria (23% in the HF group) decreased in groups supplemented with lingonberry, blackcurrant, cloudberry, sea buckthorn, and blueberry to 13%, 16%, 17%, 15%, and 16%, respectively (all *p* < 0.0001). Bacteroidetes (19% in the HF group) increased to 22% (*p* < 0.05) in the group supplemented with lingonberry and to 25% (*p* < 0.0001) in the group receiving bilberry. Verrucomicrobia (1% in the HF group) increased to 4.3% (*p* < 0.05) in the group supplemented with lingonberry and to 14% (*p* < 0.0001) in the bilberry group (Figure 6a). In the group fed with the LF diet, an increase in the relative abundance of the Bacteroidetes to 29% (*p* < 0.0001) and decreases in Firmicutes to 53% (*p* < 0.001) and Proteobacteria to 16% (*p* < 0.001) were observed when compared to the HF group (Figure 6a).

At the genus level, the principal component analysis (PCA) plot (Appendix A) revealed supplementation of the HF diet with bilberries, resulting in the most unique gut microbial profile among the groups fed HF diets. The mice fed with the LF diet had the most distinct profile in relation to other groups fed with the HF control diet or berry supplementation.

As compared to the HF control group, mice fed with the LF diet showed an increase in the relative abundances of unclassified genera from *Rikenellaceae* (*p* < 0.001) and S24–7 (*p* < 0.0001) families, and the genera *Allobaculum* (*p* < 0.0001) and *Akkermansia* (*p* < 0.05), as well as a decrease in the unclassified genera from *Lachnospiraceae* (*p* < 0.05), *Ruminococcaceae* (*p* < 0.0001), and *Desulfovibrionaceae* (*p* < 0.0001) families, and the genus *Oscillospira* (*p* < 0.0001) (Figure 6b).

Mice fed with the HF diet supplemented with lingonberries showed an increase in relative abundances of an unclassified family from *Clostridiales* (*p* < 0.0001), unclassified genus from S24–7 (*p* < 0.01), and the genus *Akkermansia* (*p* < 0.0001), as well as a decrease in unclassified genera from *Ruminococcaceae* (*p* < 0.0001) and *Desulfovibrionaceae* (*p* < 0.0001), and the genus *Oscillospira* (*p* < 0.0001), as compared to the HF control group (Figure 6b). Supplementation of the HF diet with bilberries showed an increase in an unclassified family from *Clostridiales* (*p* < 0.0001), unclassified genera from *Rikenellaceae* (*p* < 0.0001) and S24–7 (*p* < 0.001) families, and the genus *Akkermansia* (*p* < 0.0001), as well as a decrease in unclassified genera from *Ruminococcaceae* (*p* < 0.0001) and *Desulfovibrionaceae* (*p* < 0.0001), and the genus *Oscillospira* (*p* < 0.0001), as compared to the HF control group (Figure 6b). Supplementation of the HF diet with blackcurrants showed an increase in an unclassified family from *Clostridiales* (*p* < 0.0001) and unclassified genera from S24–7 (*p* < 0.05) and *Lachnospiraceae* (*p* < 0.001), and a decrease in unclassified genera from *Ruminococcaceae* (*p* < 0.0001) and *Desulfovibrionaceae* (*p* < 0.0001), as well as the genus *Prevotella2* which belongs to the family *Paraprevotellaceae* (*p* < 0.05), as compared to the HF control group (Figure 6b). Supplementation of the HF diet with cloudberries showed an increase in an unclassified family from *Clostridiales* (*p* < 0.0001) and a decrease in an unclassified genus from *Desulfovibrionaceae* (*p* < 0.0001) and the genus *Prevotella2* (*p* < 0.05), as compared to the HF control group (Figure 6b). Supplementation of the HF diet with sea buckthorn showed an increase in an unclassified family from *Clostridiales* (*p* < 0.0001), unclassified genera from S24–7 (*p* < 0.0001) and *Lachnospiraceae* (*p* < 0.0001), and the genus *Akkermansia* (*p* < 0.01) as well as a decrease in an unclassified genus from *Desulfovibrionaceae* (*p* < 0.0001), as compared to the HF control group (Figure 6b). Supplementation of the HF diet with blueberries showed an increase in an unclassified family from *Clostridiales* (*p* < 0.0001) and an unclassified genus from *Lachnospiraceae* (*p* < 0.0001), as well as a decrease in unclassified genera from *Ruminococcaceae* (*p* < 0.0001) and *Desulfovibrionaceae* (*p* < 0.0001), as compared to the HF control group (Figure 6b).

## 4. Discussion

As growing evidence reveals the effects of polyphenol-rich berries on metabolic functions, brain, and cognition [20,21,22], this study is the first to evaluate the capacity of different polyphenol-rich Nordic berries at a low dose (6% dry weight basis) in the prevention of high-fat-induced cognitive decline in mature adult C57BL/6J mice.

T-maze tests are widely used to assess spatial working memory in rodents [12,23]. Rodents with intact hippocampal functions typically display spontaneous alternation rates of around 75% or higher [12,13], meaning that their choice to explore the novel arm is based on spatial memory of the previously visited arm, and not due to random chance. In this study, the mice were around 10 months old by the time that the T-maze test was performed and the average alternation rate in the LF control group was 71%. It is possible that this impaired spatial memory performance is reflective of the ageing. HF feeding seemed to cause some further decline in performance, as there were no significant results showing that the mice in the HF control group choose to explore the novel arm any more often than due to random chance. Supplementation with the lingonberry, bilberry, blackcurrant, cloudberry, and blueberry significantly reduced the negative impact of the HF diet on spatial memory as the mice in these groups displayed average alternation rates of 73–83%. These findings are in line with a previous study in young Apoe−/− mice where animals fed with HF diets supplemented with lingonberry had spontaneous alternation rates above 90% [24]. In that study, increased hippocampal synaptic density was observed in groups supplemented with lingonberries, despite no significant differences in the T-maze alternation rate. 

A NOR test has been commonly used to measure recognition memory, attention, anxiety, and preference for novelty in rodents [25]. In this study, only the groups receiving LF diets and HF supplemented with cloudberries spent significantly more time on the Position B (novel object) in the test session compared to the time spent on the object located in the same position in the familiarization session. These results were inconsistent as compared to previous findings by Carey et al. (2014; 2021) where berry supplementation with 4% (*w*/*w*) of blueberry [7], as well as 4% (*w*/*w*) raspberry [8], could reverse the HF-driven decline in recognition memory function in C57Bl/6 male mice. In the current study, the lack of effect in relation to NOR using blueberries was somewhat unexpected, since the same blueberry (cultivar Tifblue) powder was used by Carey et al. (2014). Age differences between the mouse cohorts used, along with modest differences in the NOR protocols, could account for this variation. In addition, it has been speculated that NOR methodology might not be robust (repeatable), since previous studies [26,27] also failed to detect memory deficits in rodents using NOR. For example, Lavin et al. (2011) commented that the effects of HF diets on NOR appear to be mixed in rodents [28].

Since Nordic berries improved spatial memory performance in the T-maze, neurogenesis in dentate gyrus (DG) was investigated in a subset of animals to get a glance of potential mechanisms in action. Dietary polyphenols have been reported to modulate adult neurogenesis, especially in hippocampus [20]. DCX protein was used as a neurogenesis marker in this study, which is expressed in precursor cells and was proposed to play a role in the growth cones of neurites and in synapse formation [29]. Moreover, bilberries have been found to promote morphological modulations in hippocampal neurons, which can have a positive effect on synaptic plasticity and transmission [30].

In this study, it was shown that DCX-expressing Category 1 cells were more common for groups with LF, HF, cloudberries, and sea buckthorn diets, whereas Category 2 cells were widespread in the DG of mice fed with diets supplemented with lingonberries, bilberries, blackcurrants, and blueberries. The observation could imply that the increase in mature DCX-expressing cells with oblong processes and dendritic arborization was associated with berry supplementation, which also increases speculation that the berry supplementation could contribute to an improved neuronal plasticity in the hippocampus. Additionally, amongst all investigated berries, blackcurrant supplementation showed the most prominent changes in the quantity of DCX-expressing cells and the %DCX+ area, which has not been previously reported. Due to the low number of animal brains that could be collected for histology analysis in this study, this preliminary finding shall be considered exploratory and should be investigated further in future studies.

The observations in cell morphological changes and proliferations in this study are consistent with previous research that indicates that supplementation with blueberry, as well as polyphenol-rich extract from grape and blueberry, was associated with a more mature cell morphology profile [5,31]. A hypothesis on the effect of berries on neurogenesis may lie in the fact that the berries can improve endothelial function [32] and brain blood flow, as enhanced cerebrovascular function is known to facilitate adult neurogenesis in the hippocampus.

In our study, hippocampal levels of BDNF were not altered by the different diets. This is in contrast with previous mice studies where HF (60%E fat) control groups had significantly decreased BDNF levels in comparison with HF groups supplemented with 4% (*w*/*w*) blueberry [5]. BDNF expression may be dependent on age, diet, physical activity, and notably the brain region subjected to investigation [33,34], which may explain differences between studies. However, the fact that BDNF was not reduced by the HF diet, as compared to the LF control group, may also imply that HF did not induce a strong phenotype. 

C57Bl/6J mice fed with the HF diet are also commonly used to study the effects of a dietary intervention on the development of obesity and parameters related to the metabolic syndrome [35,36]. This aspect was also investigated, as these parameters, such as insulin resistance, may also be negatively associated with cognition [37]. Indeed, Nordic berries have previously been shown to mediate beneficial effects in this aspect, with lingonberries (at doses of 20% *w*/*w* or more) being able to prevent HF-induced weight gain, adiposity, and elevated glucose levels [24,38,39,40]. In the present study, berry supplementation did not significantly affect weight gain, fat pad weight, or glucose levels (data not shown). It is possible that the lower dose of berries (6% dwb) used here was not enough to overcome the detrimental effects of a HF diet with 60% fat, as compared to the 38–45% HF diets used in previous studies. Furthermore, the age of the mice and the timing of the dietary intervention are important to consider when designing studies and evaluating results. Younger, metabolically active mice are often used to study metabolic outcomes, whereas the more mature adult mice used in the present study deigned for cognitive outcomes. Hence, older mice may be less suitable as a model for obesity-related research questions. Nonetheless, our results show that positive effects of Nordic berry intake on cognition do not depend on body weight, as berry supplementation at the given dose did not reduce weight. Previous studies using lingonberry, blackcurrant, and bilberry supplementation have shown that reductions in weight gain can be as high as 21% in the lingonberry group compared to the HF (45%E fat) control group [40], as well as corresponding beneficial preventive effects on adiposity, liver fat, insulin resistance, and inflammatory markers [24,39]. However, low doses (4% *w*/*w*) of blueberry have been shown to not affect body weight gain when supplemented with HF diets (60%E fat) [7]. The lack of impact in this paper may be due to the usage of a lower dose of berries, the use of older animals, or the very high fat content in the diet. 

The gut and microbiota are central to many dietary effects, especially to the berry diets, as polyphenols passing through the gut can be utilized by the microbiota. Previous studies highlighted the importance of the crosstalk between the gut and the brain in metabolic control [41], while other studies suggested that the gut–brain talk in cognitive function underlies the potential mechanism of the association of metabolic syndromes with increased risk of developing cognitive impairment [3,42]. Interestingly, we found that the effects of 4-month feeding time with the HF control diet (60%E fat) induced drastic changes in the cecal microbiota of the mice, namely a reduction in Bacteroidetes and an increase in Firmicutes. This is in line with other studies which have shown that obese mice and those fed Western diets have a higher abundance of Firmicutes, concomitant with a lower abundance of Bacteroidetes [43,44]. Supplementation of the HF diet with lingonberries and bilberries showed most similar dynamic shifts, as seen in the LF group, with a significant increase in Bacteroidetes and a decrease in Firmicutes and Proteobacteria, as compared to the HF control group (Figure 6a). In these groups, there was also a notable shift in the relative abundance of the phylum Verrucomicrobia. In addition, it is commonly accepted that a healthy and resilient gut microbiome relies on high richness and biodiversity. The results of a recent study [45] also indicated an association between behavioral measures (paired-associate learning and spatial working memory) and calculated the alpha diversity of the gut microbiome in humans, where cognitive dysfunction predicted lower gut–microbiome diversity. Thus, berries that contribute to rich diversity in gut microbiota are worthy of further investigation. 

*Akkermansia muciniphila*, the main member of Verrucomicrobia dominating in the mouse and human gut, significantly increased in groups supplemented with lingonberries, bilberries, and sea buckthorn, as well as in the LF-fed group. This bacterium has previously been implicated in prebiotics-induced beneficial effects on obesity, insulin resistance, gut permeability, inflammation, and inhibited neurodegenerative processes [46,47,48]. A study on an Alzheimer’s disease mouse model has also shown that restoring this bacterium, which was depleted in APPPS1 mice as compared to healthy wild-type controls, was one of the parameters associated with reduced amyloid–beta pathology in the mouse brains [49]. Previous studies by our group and others have shown that lingonberries and other polyphenol-rich berries increase the relative abundance of *A. muciniphila*, which may be related to their beneficial metabolic anti-inflammatory effects observed in mice [39,50]. Improvements in neuroinflammatory biomarkers and synaptic density in the hippocampus have also been observed in mice fed with lingonberries, concomitant with an increase in the proportion of *A. muciniphila* [24]. Although no significant effects or correlations between *A. muciniphila* and the investigated metabolic and cognitive biomarkers were observed in this study, a drastic increase in the proportion of *Akkermansia* following the consumption of lingonberries, bilberries, and sea buckthorn, despite the low dose (6% dry weight basis), may be worth investigating further.

In conclusion, we identified novel berry species of interest as foods promoting brain health. The undertaken study shows that lingonberry, bilberry, blackcurrant, blueberry, and cloudberry influence spatial memory in a positive manner and may confer beneficial actions on hippocampal neurogenesis and gut microbiome diversity. This study suggests that Nordic berries should be further considered to identify dietary approaches and counteract age-related changes in the brain and overall brain health.

## Figures and Tables

**Figure 1 nutrients-14-02734-f001:**
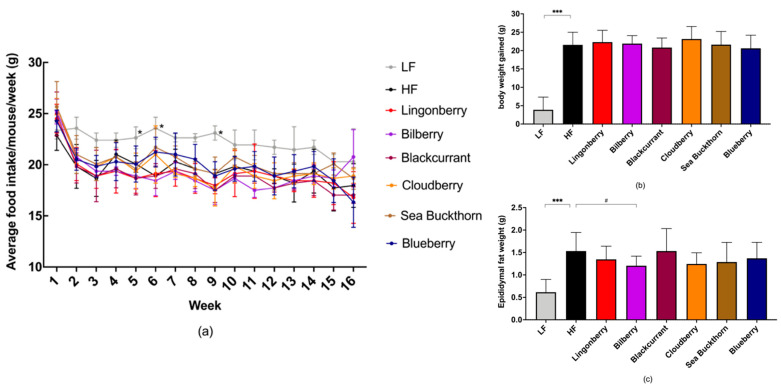
The effect of HF and berry supplementation on food intake, body weight, and epididymal fat pad weight. (**a**) The graph depicts mean weekly food intake in the different diet groups, measured per cage (3–4 cages per diet group); (**b**) body weight gain; (**c**) epidydimal fat pad weight at the end of the study. One-way ANOVA was used for data analyses and Dunnett’s multiple comparisons test was used for post-hoc analysis to compare each diet to the HF diet. Significant differences denoted by * *p* < 0.05, *** *p* < 0.001. (# *p* < 0.1 denotes a non-significant trend). Values are represented as mean ± SD for *n* = 14–15 per group.

**Figure 2 nutrients-14-02734-f002:**
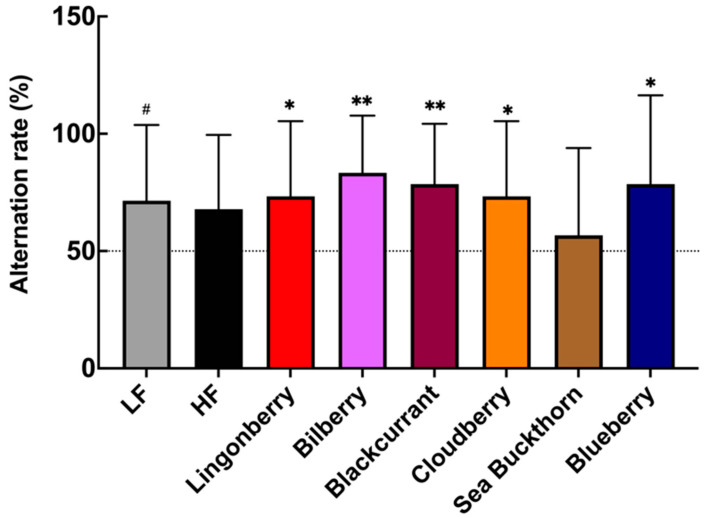
The effects of the HF diet and berry supplementation on the alternation rate in a T-maze test. Each bar represents the mean alternation rate of the mice in each diet group. The Wilcoxon signed-rank test was applied to compare the % alternation of each group against a 50% chance. * *p* < 0.05, ** *p* < 0.01. (# *p* < 0.1 denotes a non-significant trend). Data represent mean ± SD for *n* = 14–15 per group.

**Figure 3 nutrients-14-02734-f003:**
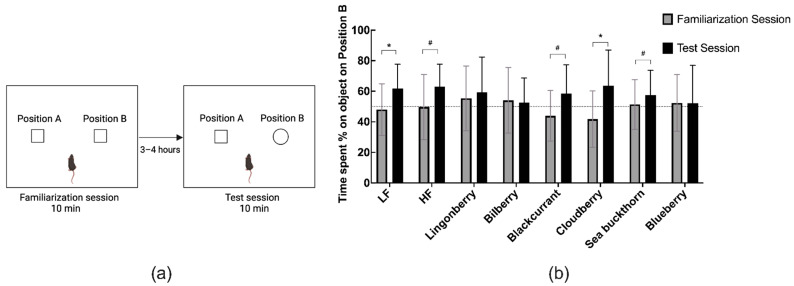
(**a**) Novel object recognition (NOR) experimental scheme. (**b**) Percentage time spent exploring the object on Position B in the familiarization and test sessions. Values are represented as mean ± SD (*n* = 14–15). The results in the familiarization session were compared to the test session using the non-parametric Wilcoxon signed-rank test. * *p* < 0.05. (# *p* < 0.1 denotes a non-significant trend).

**Figure 4 nutrients-14-02734-f004:**
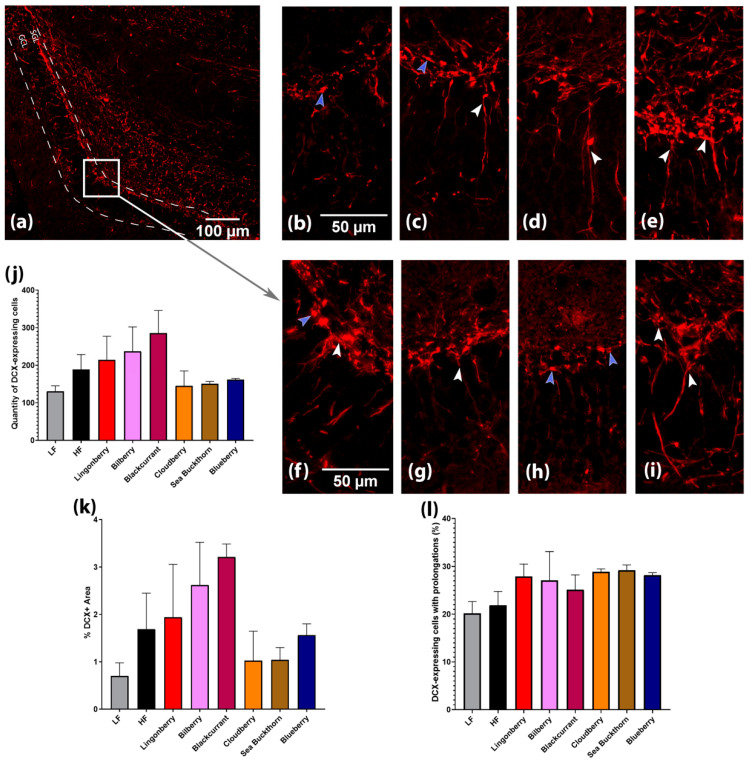
Hippocampal neurogenesis in mice fed with different diets. (**a**–**i**) Images visualized the localization of DCX-expressing cells in the sub-granular layer (SGL) and its prolongations in the granule cell layer (GCL) of dentate gyrus (DG). (**a**) DG at ×100 magnification. (**b**–**i**) DG at ×400 magnification of mice fed different diets: (**b**) LF diet; (**c**) HF diet; (**d**) HF + lingonberry; (**e**) HF + bilberry; (**a**,**f**) HF + blackcurrant; (**g**) HF + cloudberry; (**h**) HF + sea buckthorn; (**i**) HF + blueberry. White arrowheads mark DCX-expressing cells with prolongations, and purple arrowheads show cells without prolongations. (**j**) Quantification of DCX-expressing cells in DG; (**k**) quantification of %DCX+ Area; and (**l**) quantification of DCX-expressing cells with prolongations (%). Values are represented for slides from *n* = 2–3 brains per diet group.

**Figure 5 nutrients-14-02734-f005:**
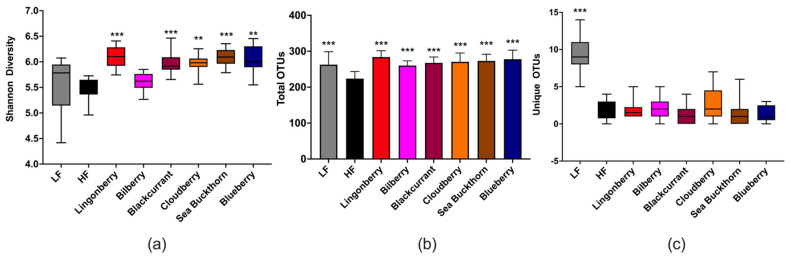
Alpha diversity of the gut microbiota in mice fed LF, HF, lingonberry, bilberry, blackcurrant, cloudberry, sea buckthorn, and blueberry (*n* = 12–15 per group) measured by (**a**) Shannon diversity index; (**b**) total OTU richness; and (**c**) unique observed species. The Shannon index compared each group to the HF control group using the Kruskal–Wallis rank-sum test followed by pairwise comparisons using the Wilcoxon rank-sum test. The unique observed species and the total OTU richness were compared using the ANOVA test followed by the least significant difference (LSD) post-hoc test when ANOVA indicated significance. ** *p* < 0.01, *** *p* < 0.001.

**Figure 6 nutrients-14-02734-f006:**
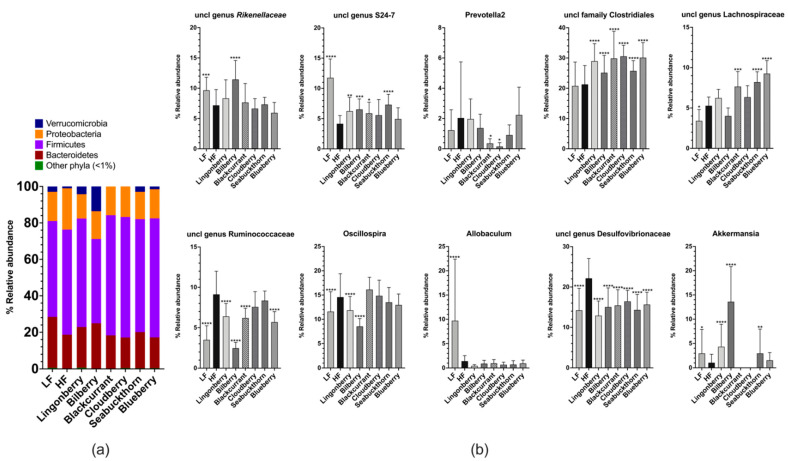
Composition of the gut microbiota: (**a**) phylum level (data represent mean values); (**b**) genus levels in the cecum of mice fed the experimental diets for 16 weeks (*n* = 12–15 per group) represent mean ± SD. Each group was compared to the HF control group using two-way ANOVA and the *p*-values were corrected for multiple comparisons by controlling the false discovery rate (FDR) using the original FDR method of Benjamini and Hochberg. * *p* < 0.05, ** *p* < 0.01, *** *p* < 0.001, **** *p* < 0.0001.

## Data Availability

The data underlying this study are available in the published article and its online Appendix A.

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
