# Peer review of "Identification of Nordic Berries with Beneficial Effects on Cognitive Outcomes and Gut Microbiota in High-Fat-Fed Middle-Aged C57BL/6J Mice"

_nutrients, 2022, doi:10.3390/nu14132734_

Round 1

Reviewer 1 Report

In the present manuscript Fang Huang and colleagues investigated the potential benefits of a diet supplemented with different types of berries on cognitive impairment and gut microbiota in mature-adult mice. The authors have described that high fat-berry supplementation-fed mice showed improved spatial recognition as measured in the T maze alternation test and greater proportion of doublecortin -expressing cells in the hippocampus compared to animals fed with a low-fat diet. Moreover, the microbiota found in those animals was increased by 4-14 times. The authors conclude a potential benefit of eating berries to preserve memory function and to induce alteration in the gut microbiota.

Collectively, the manuscript is well written and presented. However, the experimental design and analysis of the data obtained refrain me to completely agree with the conclusions of the work.

Major

·       Is not the experiment lacking a normal chow-fed group?

·       Is there any “waiting” time between the 1st and 2nd phase in the t-maze spontaneuos alternation?

·       What is the rationale to analyse the data of the novel recognition test that way? Usually, it is calculated as a discrimination index computed as the time exploring the new object divided by the total time exploring. Healthy rats usually spend more time exploring the novel, so a reduction in this discrimination index indicated a deficit in recognition memory.

·       It is not clear to me if comparing the alternation rate in the T-maze against the 50% (chance) is a correct analysis. Looking at the graph it looks like that compared to the high-fat diet group, the berrie-supplemented diet seems to not induce any effect. Furthermore, it also seems that there are no differences between low and high- fat diet in this test, so it is a bit tough to me to conclude that the berries are somehow improving cognitive cognition at least as the data are analysed here.

·       Similar doubts regarding the analysis of the novel recognition task. Have the authors calculated the discrimination index as it has been previously published (i.e Nutrients 2020 May 23;12(5):1520. doi: 10.3390/nu12051520). The comparison in this test should be done between different exploration times within the same test session. Comparing exploration times from different session seem not to be the best approach. Again, looking at the graph and focusing only in the test session data it seems that there are no statistical differences between the groups, but it would be worth to try the analysis proposed.

Minor:

·       Line 41: replace “treat” by “prevent”.

·       Line 47: “within the area” replace by “in the field of”

·       Line 54: add (HF) after high-fat feeding

·       Typo in line 128 “alternation”.

·       In general, the figures seem to me very small, especially the figure 4 is very difficult to read. I would increase the size of the figure and font. Also, the immunofluorescence images are small and too dark, and it is difficult to observe any difference between the groups.

Reviewer 2 Report

1. In methods, page 4, 2.4. Why were the animals fasted 4h before tissue and blood samples were collected?

2. In methods, page 4, 2.5. The sample size used in the histology studies (staining of DCX-expressing neurons in the hippocampus) is small (n = 2-3 animals per group). The authors state they use 14-15 animals in each diet group. Thus, it is unclear why only 2-3 animals were used for the histology studies. I am afraid this does not confer enough power to make a valid analysis.

3. In methods, page 5, line 8, 2.7. Define the abbreviation "DCX" at its first appearance.

4. Figure 4 legend. There is no explanation for (a). In addition, the black arrowheads are hard to visualize. Images (j,k,l) are hard to see. Is there any significant difference among different diets?

5. Figure 5 legend. What are the n numbers for each group?

6. In discussion, page 14, line 5, metabolic syndromes (MetS) appears only once in this manuscript, thus, there is no need to define the abbreviation.

Round 2

Reviewer 1 Report

The authors investigated the potential benefits of a diet supplemented with different types of berries on cognitive impairment and gut microbiota in mature-adult mice.

After the second submission, the paper has improved significantly. The authors have incorporated throughout the text information that was relevant and was missing in the first version.

The authors have answered my previous comments satisfactorily. Yet, I would only suggest increasing the size of figure 6 as it is really difficult to see.

Author Response

Manuscript ID: nutrients-1735617

We thank the reviewer for the prompt reply and valuable comments. Regarding the comment in Figure 6, improvements are made accordingly. 

Best regards,

Fang Huang

25 June 2022, Lund

Reviewer 2 Report

All my comments have been addressed adequately by the authors. I have no further comments.

Author Response

On behalf of all co-authors, I would like to thank the referee for the valuable time to review our manuscript. 

Best regards,

Fang Huang

25 June 2022, Lund